# A Retrospective Analysis of Intravenous Push versus Extended Infusion Meropenem in Critically Ill Patients

**DOI:** 10.3390/antibiotics13090835

**Published:** 2024-09-02

**Authors:** Emory G. Johnson, Kayla Maki Ortiz, David T. Adams, Satwinder Kaur, Andrew C. Faust, Hui Yang, Carlos A. Alvarez, Ronald G. Hall

**Affiliations:** 1Texas Health Presbyterian Hospital Dallas, Dallas, TX 75231, USAandrewfaust@texashealth.org (A.C.F.); 2Health Sciences Center, Jerry H. Hodge School of Pharmacy, Texas Tech University, Dallas, TX 75235, USAcarlos.alvarez@ttuhsc.edu (C.A.A.); ronald.hall@ttuhsc.edu (R.G.H.2nd); 3Texas Health Harris Methodist Hospital Fort Worth, Fort Worth, TX 76104, USAsatwinderkaur@texashealth.org (S.K.); 4Center for Real-World Evidence, Dallas, TX 75235, USA

**Keywords:** meropenem, extended infusion, intravenous push, critical illness, anti-bacterial agents, intensive care units, hospitals, drug administration

## Abstract

Meropenem is a broad-spectrum antibiotic used for the treatment of multi-drug-resistant infections. Due to its pharmacokinetic profile, meropenem’s activity is optimized by maintaining a specific time the serum concentration remains above the minimum inhibitory concentration (MIC) via extended infusion (EI), continuous infusion, or intermittent infusion dosing strategies. The available literature varies regarding the superiority of these dosing strategies. This study’s primary objective was to determine the difference in time to clinical stabilization between intravenous push (IVP) and EI administration. We performed a retrospective pilot cohort study of 100 critically ill patients who received meropenem by IVP (*n* = 50) or EI (*n* = 50) during their intensive care unit (ICU) admission. There was no statistically significant difference in the overall achievement of clinical stabilization between IVP and EI (48% vs. 44%, *p* = 0.17). However, the median time to clinical stability was shorter for the EI group (20.4 vs. 66.2 h, *p* = 0.01). EI administration was associated with shorter hospital (13 vs. 17 days; *p* = 0.05) and ICU (6 vs. 9 days; *p* = 0.02) lengths of stay. Although we did not find a statistically significant difference in the overall time to clinical stabilization, the results of this pilot study suggest that EI administration may produce quicker clinical resolutions than IVP.

## 1. Introduction

Meropenem is a broad-spectrum beta-lactam antibiotic that belongs to the carbapenem class. Meropenem is commonly used in critically ill patients who are at risk of infection with multi-drug-resistant pathogens. Beta-lactam antibiotics have shown greater efficacy in vitro against bacteria when the percent of time during the dosing interval spent in excess of their minimum inhibitory concentration (MIC), often referred to a % T > MIC, is increased [1,2,3]. Traditionally, beta-lactams are dosed intermittently; however, this necessitates frequent dosing to overcome the short half-life of beta-lactams and maintain an adequate % T > MIC [1,4,5,6]. In recent years, the use of continuous and extended infusion (EI) administration has been more widely adopted with beta-lactams to achieve concentrations above the MIC for longer periods of time, which should result in optimal antimicrobial activity [7,8]. The use of EI has subsequently been adopted by the 2021 Surviving Sepsis Campaign as the recommended dosing strategy for beta-lactam antibiotics over bolus infusions [9].

The current literature is conflicting regarding the superiority of EI versus intermittent dosing, especially with meropenem in critically ill patients. Few randomized controlled trials have demonstrated the superiority of extended infusion over intermittent dosing [8,10]. Meta-analyses, including of meropenem in patients with sepsis or severe infections, have also reported that extended or continuous infusions were associated with improvements in several clinical outcomes, including mortality and clinical cure rates [11,12,13]. However, more recent clinical trials have failed to demonstrate a statistically significant reduction in 90-day mortality with the use of EI meropenem [1,14,15]. Thus, the clinical superiority of one administration technique over another is uncertain.

After Hurricane Maria devastated the production of IV fluids in Puerto Rico, the IVP route was initiated in our hospital system. At that point, IV meropenem was transitioned from the IVPB (over 30 min) to the IVP (over 5 min) route of administration. As the shortage resolved, due to increasing amounts of research in support of EI administration [7,8,9,10,11,12,13], the decision was made to begin the preferential use of EI meropenem (over 180 min). However, as there are no clear data that support EI over IVP, we conducted a retrospective analysis to determine any differences between these administration techniques.

## 2. Results

### 2.1. Patients

A total of 449 patients between January 2023 and November 2023 were screened for study inclusion. A total of 50 patients were included in each study arm, 308 patients were excluded, and 41 patients were not randomly selected as part of the 100-patient pilot (Figure 1). 

The baseline characteristics between the two groups were, overall, well balanced (Table 1). The average age between IVP and EI was 61 and 67, respectively. The majority of our patient population in each group was white (72% and 84%). About one-third of patients had some form of antibiotic administration (IV or oral) within 90 days of their current admission. Another third of patients had a previous hospital admission within the 90 days prior to their current admission. The most common documented source of infection was respiratory (42% and 48%). There was a significant difference in the median time from admission to the first dose of meropenem between IVP and EI (23 h [IQR 5–194] vs. 119 h [IQR 20–372], *p*-value = 0.003). There was also a significant difference in patients with an elevated respiratory rate at the start of meropenem between IVP and EI (40 vs. 47, *p*-value = 0.04). Meropenem Q8 h was the most frequently used dosing interval for both groups. However, the EI group used Q8 h more than the IVP, while Q12 h dosing was used more in the IVP group. IVP patients also trended toward being more likely to require non-invasive oxygen (nasal cannula, BiPAP, or CPAP) and having no bacterial organism cultured, but these were not statistically significant.

### 2.2. Primary Outcome

Clinical stabilization was only achieved in 24 (48%) patients in the IVP group and 22 (44%) patients in the EI group (*p* = 0.84) (Table 2). The separation shown in the Kaplan–Meier curve also shows a lack of significance in the overall time to clinical stabilization; *p* = 0.1710 (Figure 2). The Cox proportional hazards model yielded similar results (Hazard Ratio 1.52, 95% Confidence Interval 0.84–2.77) when adjusting for age, time to first dose, Charlson comorbidity index score, the use of mechanical ventilation, and vasopressor use. However, there was a statistically significant difference in the median time to clinical stabilization in the EI group compared to the IVP group (20.4 h vs. 66.2 h, *p*-value = 0.01).

### 2.3. Secondary Outcomes

Secondary outcomes are displayed in Table 3. There were no significant differences in the time to defervescence or the normalization of the WBC. Hospital length of stay (13.0 days [IQR 6.22–20.13] vs. 16.9 days [IQR 12.16–29.92], *p*-value = 0.05) and ICU length of stay (6.1 h [IQR 2.97–8.39] vs. 9.2 h [IQR 4.67–15.71], *p*-value = 0.02) were significantly shorter in the EI group compared to the IVP group, respectively. There were no statistically significant differences between the IVP and EI group in terms of mortality, treatment failure, microbiological cure, the recurrence of resistant isolates, or isolates specifically resistant to meropenem (Table 3). 

Lastly, there was also no statistically significant difference regarding the rates of culture growth or lack thereof. In patients that did have a positive culture, there were significantly more patients that grew *Escherichia coli* in the EI group compared to the IVP group (1 vs. 8, *p*-value = 0.03) (Table 4).

## 3. Discussion

The pilot study was unable to observe a statistically significant difference between IVP and EI meropenem regarding the percentage of patients achieving clinical stabilization or the overall time to clinical stabilization. This lack of statistical significance for the overall time to clinical stabilization is likely caused by the small sample size of our pilot cohort. However, there was a statistically significant difference in the median time to clinical stability, which could point to a potential benefit of EI over IVP in this patient population. This could be due to the longer % T > MIC achieved with EI dosing, as mentioned previously, which in turn has been shown to have better bacteriostatic and bactericidal properties [3,14], leading to reductions in bacterial burden [16] and lengths of stay, especially in the ICU [14,17].

Various meta-analyses have identified mortality benefits with other beta-lactam antibiotics [17,18,19,20], with a recent retrospective cohort study identifying a mortality benefit with EI beta-lactam antibiotics, with the most benefit seen in patients that were critically ill or infected with organisms with elevated MICs [21]. Our study found no difference in mortality rates, which agrees with the outcome found in the MERCY trial, which observed a 42% rate of all-cause mortality at 90 days in both its groups [1]. Likewise, the recently published 7202 patient BLING-3 trial only observed a 1.9% decrease in 90-day mortality and failed to observe a statistically significant difference [15]. This trend can also be seen in other trials comparing intermittent vs. extended infusion meropenem [14] and IVP vs. intermittent infusion [22]. However, the opposite can be seen in a meta-analysis by Chen and colleagues, which found a 34% lower risk of mortality with extended over intermittent infusions that was statistically significant [11]. This can also be seen in another meta-analysis that was heavily dependent on BLING-3, which observed a 2.5% decrease in 90-day mortality that was statistically significant [12]. BLING-3 and its accompanying meta-analysis similarly show that achieving a large mortality difference is difficult even with a sample size of several thousand patients. Lastly, our findings that EI was associated with significantly lower hospital and ICU lengths of stay agree with Tran and colleagues’ comparison of intermittent vs. EI beta-lactam antibiotics [14].

Since no significant difference in the percentage of patients achieving clinical stabilization was found between IVP and EI, one can conclude that either option is viable. However, that may be short-sighted given that EI may be more favorable due to its quicker median time to clinical stability and shorter lengths of ICU and hospital stay in survivors. IVP may still remain useful in select patient populations. For example, since IV medications contribute to 61% of a patient’s fluid intake on day one of their ICU admission and 40% during their first seven days, a lower-volume dosing strategy may be more beneficial in patients that are fluid-restricted [23]. IVP may also be superior in achieving quicker administration rates for first dose antibiotics in septic patients within an emergency medicine setting [24,25].

## 4. Limitations

There were several limitations to our pilot study, the first being that it is retrospective in nature. However, the study allows for the evaluation of these two types of administrations in a real-world setting.

Second, our study included a small number of patients, which likely limited our ability to detect a statistically significant difference in the overall time to clinical stability. 

Third, there was a numerically larger difference in time to first dose between the IVP and EI group. We suspect this difference was due to a transition to EI after being on an alternative antibiotic regimen initially. However, we adjusted for this difference in the Cox proportional hazards model. There was also a numerically higher number of patients receiving invasive oxygen at the start of meropenem initiation in the EI compared to the IVP group, which was also adjusted for using the Cox proportional hazards model. 

Lastly, the frequency at which lab values, vital signs, and microbiological cultures were reported could have impacted our ability to accurately determine a time to clinical stabilization, especially in terms of WBC monitoring. We noticed that the return to a stable WBC level, as deemed by the SIRS criteria, often remained the final criteria needed to classify a patient as clinically stable. Since our hospital’s practice is to measure WBCs once per day with morning labs, the determination of clinical stability could be limited by this protocol. Therefore, more frequent WBC monitoring or differing hospital protocols may yield shorter times to clinical stabilization.

## 5. Materials and Methods

### 5.1. Study Design

We performed a retrospective cohort study in two hospitals within the Texas Health Resources hospital system: Texas Health Presbyterian Hospital Dallas (THD) and Texas Health Harris Methodist Hospital Fort Worth (THFW). The time period selected for the study was January 2023 to October 2023.

Patients were categorized as IVP or EI based on the type of administration method the patient received for the majority of their hospital admission. IVP was defined as meropenem administration over 5 min, while EI was defined as meropenem administration over 180 min (3 h). Patients were included in the study if they were 18 years of age or older, admitted into the ICU, treated with IVP or EI meropenem for at least 48 h during their ICU stay, and had a positive systemic inflammatory response syndrome (SIRS) criteria score at the time of meropenem initiation. The SIRS criteria score consists of 4 components: a temperature >38 °C (100.4 °F) or <36 °C (96.8 °F), a heart rate >90 beats per minute, a respiratory rate >20 breaths per minute, and a white blood cell count (WBC) >12,000/mm^3^ or <4000/mm^3^ [26]. A positive SIRS score was defined as meeting ≥2 criteria. A patient was defined as critically ill if they had a positive SIRS score accompanied with an ICU admission for further clinical management.

### 5.2. Patient Selection

Patients were excluded if they were less than 18 years of age, met fewer than 2 SIRS criteria at the start of meropenem treatment, were transitioned to hospice or comfort care within 48 h of admission, died within 48 h of admission, received IVP or EI meropenem for less than 48 h, or received meropenem as an intermittent infusion for over 30 min for the majority of their admission.

Patient data were collected from electronic medical records at the study institutions. Patients were divided into two categories, IVP and EI, based on the administration type the patient received for the majority of their hospital stay. Within each group, patients were assigned a trial number. A random number generator was used for each group to randomly select patients for inclusion or exclusion. Selection for the pilot study within each group ceased once 50 patients had met the inclusion criteria in each arm. The investigators estimated that approximately 50 patients from the IV push arm would meet the eligibility criteria due to the period of usage of the IV push route.

### 5.3. Primary and Secondary Outcomes

The primary outcome was the time to clinical stabilization. Clinical stabilization was defined as the resolution of all SIRS criteria that were present at the initiation of meropenem.

The secondary outcomes were the percentage of patients who achieved clinical stabilization, ICU length of stay, hospital length of stay, time to complete defervescence from the start of meropenem, time to WBC normalization from the start of meropenem, treatment failure, microbiologic cure, the recurrence of resistant isolates, isolates resistant to meropenem, and mortality. Time to complete defervescence was defined as the point at which the patient was afebrile (temperature < 38 °C (100.4 °F) or >36 °C (96.8 °F)), with no recurrent febrile events for the remainder of their hospital stay. WBC normalization was defined as the point at which the WBC remained less than 12,000/mm^3^ and greater than 4000/mm³ with no further elevation or decline outside this range for the remainder of their hospital stay. Microbiological cure was defined as the clearance of bacteria on a repeated culture. The recurrence of resistant isolates was defined as a repeat positive culture for the same bacteria, which were resistant to ≥2 antibiotic classes, as in the initial culture. Isolates were defined as resistant to meropenem based on The Clinical & Laboratory Standards Institute (CLSI) breakpoint criteria. Lastly, treatment failure was defined as the failure to resolve all SIRS criteria present at the start of meropenem.

### 5.4. Statistical Analysis

In this pilot study, we randomly chose 100 patients who received meropenem by IVP or EI. Categorical data were analyzed using a chi-squared test or Fisher’s exact test as appropriate. Continuous data were analyzed using the Wilcoxon Rank-Sum test or Student *t*-test as appropriate. Our primary outcome, the time to clinical stabilization, was described using a Kaplan–Meier curve and analyzed with a log-rank test. A Cox proportional hazard model was also used to assess covariates. A censor was applied to the Kaplan–Meier curve if patients required more than 10 days of meropenem, were discharged, or did not achieve clinical stabilization. A *p*-value < 0.05 was chosen to determine statistical significance. Statistical analyses were conducted using Stata version 15.1 (StataCorp, College Station, TX, United States) and SAS 9.4 (SAS Institute Inc., Cary, NC, United States).

## 6. Conclusions

Based on the available evidence, there is no significant difference between IVP and EI regarding the percentage of patients who achieved clinical stabilization and their overall time to clinical stabilization. However, patients within the EI group did have a significantly shorter median time to clinical stabilization, as well as shorter median hospital and ICU lengths of stay compared to the IVP group. Therefore, we conditionally recommend using EI over IVP when possible. Well-designed randomized controlled trials could provide additional clarity regarding the impact of these administration strategies on meropenem’s clinical effectiveness.

## Figures and Tables

**Figure 1 antibiotics-13-00835-f001:**
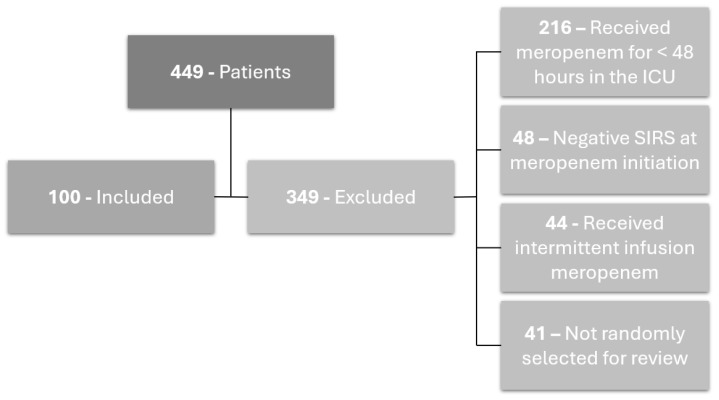
Flow chart of patients included and reasons for exclusion of patients.

**Figure 2 antibiotics-13-00835-f002:**
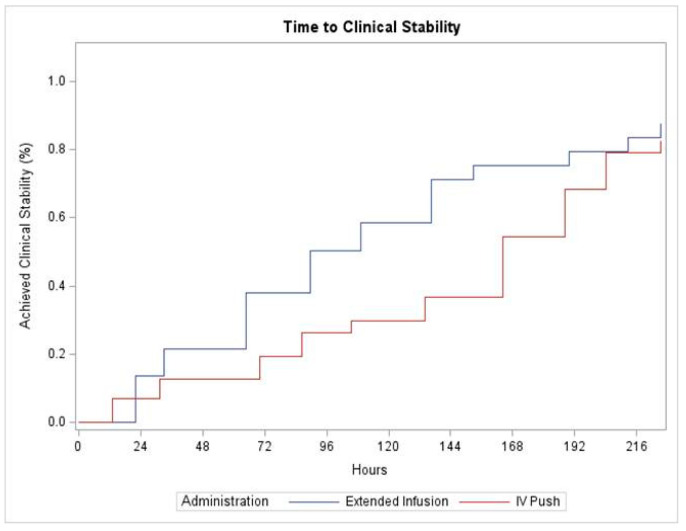
Kaplan–Meier curve showing the time to clinical stabilization.

**Table 1 antibiotics-13-00835-t001:** Baseline characteristics.

Characteristics	Intravenous Push (*n* = 50)	Extended Infusion (*n* = 50)	*p*-Value
Age (years), median (IQR)	61 (49–73)	66 (53–72)	0.57
Male sex, *n* (%)	24 (48)	32 (64)	0.11
Race, *n* (%) White Black Asian More than one Declined to answer/unknown	36 (72) 11 (22) 0 (0) 1 (2) 2 (4)	42 (84) 6 (12) 1 (2) 0 (0) 1 (2)	0.39
Hispanic or Latino, *n* (%)	4 (8)	8 (16)	0.36
Weight (kg), median (IQR)	82 (67–96)	82 (66–95)	0.89
Height (inches), median (IQR)	67 (64–70)	67 (65–71)	0.49
Body mass index (kg/m^2^), median (IQR)	27 (22–34)	26 (24–30)	0.63
Creatinine clearance (mL/min), median (IQR)	62 (28–86)	68 (32–95)	0.46
Antibiotic therapy 3 months prior to admission, *n* (%)	14 (28)	17 (34)	0.52
Hospitalization 3 months prior to admission, *n* (%)	16 (32)	18 (36)	0.67
Charlson Comorbidity Index, median (IQR)	4 (1–5)	4 (2–5)	0.96
Oxygen status, *n* (%) Room air Non-invasive oxygen Invasive oxygen	5 (10) 24 (48) 21 (42)	1 (2) 17 (34) 32 (64)	0.06
Vasopressor use, *n* (%)	34 (68)	30 (60)	0.41
Time from admission to first dose of meropenem (hours), median (IQR)	23 (5–194)	119 (20–372)	0.003
Febrile at meropenem start, *n* (%)	22 (44)	23 (46)	0.84
Abnormal WBC at meropenem start, *n* (%)	45 (90)	42 (84)	0.37
Elevated respiratory rate at meropenem start, *n* (%)	40 (80)	47 (94)	0.04
Elevated heart rate at meropenem start, *n* (%)	42 (84)	45 (90)	0.37
Meropenem dose, *n* (%) * 500 mg 1000 mg 2000 mg	8 (16) 41 (82) 1 (2)	2 (4) 45 (90) 3 (6)	0.09 0.39 0.62
Frequency, *n* (%) * Q6 h Q8 h Q12 h Q24 h	2 (4) 28 (56) 19 (38) 1 (2)	0 (0) 38 (76) 9 (18) 3 (6)	0.49 0.03 0.03 0.62
Appropriate renal adjustment, *n* (%)	49 (98)	48 (96)	1.00
Primary source of infection, *n* (%) Respiratory tract Gastrointestinal tract/intra-abdominal Genitourinary tract Bloodstream Skin and soft tissue Central nervous system Other	21 (42) 12 (24) 5 (10) 4 (8) 5 (10) 1 (2) 2 (4)	24 (48) 7 (14) 4 (8) 6 (12) 3 (6) 6 (12) 0 (0)	0.26

*n* = number; IQR = interquartile range; kg = kilograms; m = meter; mL = milliliter; min = minute; mg = milligram; Q = every; h = hours; * The dose and frequency noted were based on the dose used for the majority of the patient’s admission.

**Table 2 antibiotics-13-00835-t002:** Percent of patients achieving clinical stabilization.

Outcome	IVP (*n* = 50)	EI (*n* = 50)	*p*-Value
Achieved Clinical Stabilization, *n* (%)	24 (48)	22 (44)	0.84
Median Time to Clinical Stabilization (hours), median (IQR)	66.2 (14.4–177.3)	20.4 (73.8–89.7)	0.01

**Table 3 antibiotics-13-00835-t003:** Secondary outcomes.

Outcomes	IVP Group (*n* = 50)	EI Group (*n* = 50)	*p*-Value
Time to complete defervescence (hours), median (IQR) *	26.14 (13.98–61.21)	14.78 (9.41–35.17)	0.29
Time to WBC normalization (hours), median (IQR) **	127.25 (77.45–166.98)	70.63 (33.94–136.98)	0.08
Hospital length of stay (days), median (IQR) ***	16.92 (12.16–29.92)	13.03 (6.22–20.13)	0.05
ICU length of stay (days), median (IQR) ***	9.19 (4.67–15.71)	6.07 (2.97–8.39)	0.02
Mortality, *n* (%)	13 (26)	19 (38)	0.20
Treatment failure, *n* (%)	19 (38)	25 (50)	0.23
Microbiological cure, *n* (%)	12 (24)	6 (12)	0.14
Recurrence of resistant isolates, *n* (%)	3 (6)	2 (4)	1.00
Isolates resistant to meropenem, *n* (%)	1 (2)	1 (2)	1.00

* *n* = 22; patients not febrile at meropenem start were excluded ** *n* = 45; patients without abnormal WBC at meropenem start were excluded *** *n* = 37; inpatient mortality was excluded.

**Table 4 antibiotics-13-00835-t004:** Microbiological characteristics.

Characteristics	Intravenous Push (*n* = 50)	Extended Infusion (*n* = 50)	*p*-Value
No pathogen cultured, *n* (%)	24 (48)	15 (30)	0.07
Organism(s) cultured, *n* (%) *Acinetobacter* spp. *Klebsiella* spp. *Pseudomonas* spp. *Escherichia coli* *Enterococcus* spp. *Staphylococcus* spp. *Streptococcus* spp. *Proteus* spp. *Candida* Other	0 (0) 1 (2) 3 (6) 1 (2) 3 (6) 8 (16) 2 (4) 1 (2) 4 (8) 6 (12)	2 (4) 5 (10) 8 (16) 8 (16) 4 (8) 9 (18) 3 (6) 2 (4) 1 (2) 10 (20)	0.50 0.20 0.20 0.03 1.00 1.00 1.00 1.00 0.37 0.41

*n* = number; spp. = species.

## Data Availability

The data that support the findings of this study are available from Texas Health Resources. Restrictions apply to the availability of these data, which were used under license for this study. Data are available from the authors with the permission of Texas Health Resources.

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
