# Peer review of "A Retrospective Analysis of Intravenous Push versus Extended Infusion Meropenem in Critically Ill Patients"

_antibiotics, 2024, doi:10.3390/antibiotics13090835_

Round 1

Reviewer 1 Report

Comments and Suggestions for Authors

The Manuscript antibiotics-3146218-peer-review-v1

Manuscript antibiotics-3146218-peer-review-v1 describes the impact of method of meropenem administration on the therapeutic outcome in critically ill patients.

The manuscript is very well-written and within the scope of Antibiotics. The manuscript discussed a clinically-important topic that could significantly affect the therapeutic outcome of meropenem treatment.

Major concerns

1.       Authors need to describe how sample size was determined at the stage of experimental design. Of particular interest, what was the rationale of ceasing patients’ collection one 50 patients had met the inclusion criteria?

2.       Discussion: Authors need to discuss the effect of method of administration on the pharmacokinetic profile (plasma level variations) of meropenem and subsequent pharmacodynamic effects on median time to clinical stabilization and hospital stay.

Author Response

1.) Authors need to describe how sample size was determined at the stage of experimental design. Of particular interest, what was the rationale of ceasing patients’ collection one 50 patients had met the inclusion criteria?

Thank you for your feedback, we have included more information on page 2, lines 88-91 on the manuscript:

"Selection for the pilot study within each group ceased once 50 patients had met the inclusion criteria in each arm. The investigators estimated approximately 50 patients from the IV push arm would meet eligibility criteria due to the period of usage of the IV push route."

2)  Discussion: Authors need to discuss the effect of method of administration on the pharmacokinetic profile (plasma level variations) of meropenem and subsequent pharmacodynamic effects on median time to clinical stabilization and hospital stay.

Thank you for the feedback, we have expanded more on this within our discussion section, highlighting connections with other studies we came across on page 6, lines 190-194:

"This could be due to the longer % T>MIC achieved with EI infusion dosing, as mentioned previously, which in turn has been shown to denote better bacteriostatic and bactericidal properties [3,14], reductions in bacterial burden [17], and lengths of stay, especially ICU [14,18]."

Reviewer 2 Report

Comments and Suggestions for Authors

1- The language of the article is good.

2- Abstract: well written

3- Introduction:

3.1. Please give more information about the current strategies of beta-lactam infusion (lines 50-55). Also, add relevant references.

4- Material and method:

4.1. Study design: Please add the patient number and information about gender, age, etc. Briefly add relevant sentences and fully describe the study design.

4.2. Did you receive ethical permission from two hospitals separately? Please indicate it in the date and ethical permission number order.

4.3. Please move lines 75-79 to the 2.3 patient selection part.

5- Discussion:

5.1. Please add the limitation of the study as a section at the end of the Discussion.

5.2. Please add some sentences to support why shorter median hospital lengths of stay

was seen in the EI group in comparison to the IVP group.

5.3. Please indicate it also about ICU lengths of stay.

Author Response

3- Introduction:

3.1. Please give more information about the current strategies of beta-lactam infusion (lines 50-55). Also, add relevant references.

Thank you for the feedback, we have included our hospital's current strategy for beta-lactam infusion on page 2, lines 52-58:

"After Hurricane Maria devastated the production of IV fluids in Puerto Rico, IVP route was initiated in our hospital system. At that point IV meropenem was transitioned from the IVPB (over 30 minutes) to the IVP (over 5 minutes) route of administration. As the shortage resolved, due to increasing literature to support EI administration [7-13], the decision was made to begin the preferential use of EI meropenem (over 180 minutes). However, as there is no clear data to support EI over IVP, we conducted a retrospective analysis to determine any difference between administration techniques."

4- Material and method:

4.1. Study design: Please add the patient number and information about gender, age, etc. Briefly add relevant sentences and fully describe the study design.

Thank you for your feedback, we are unsure on what the reviewer means by this comment since we include both age and biological sex in the baseline characteristics, which can be found in table 1 on page 4. We are also unsure on what changes need to be made to fully describe the study design, we would appreciate the reviewer providing specific examples of their expectations.

4.2. Did you receive ethical permission from two hospitals separately? Please indicate it in the date and ethical permission number order.

Thank you for your comment, the study was approved by the local institutional review board. Information can be found in the IRB statement section on page 8, lines 259-261:

"Institutional Review Board Statement: The study was conducted in accordance with the Declaration of Helsinki and approved by the Institutional Review Board of UT Southwestern (protocol code STU-2023-1074) on November 13, 2023."

4.3. Please move lines 75-79 to the 2.3 patient selection part.

Lines have been moved and are now found on page 2, line 79 - 83

5- Discussion:

5.1. Please add the limitation of the study as a section at the end of the Discussion.

Thank you for the feedback, in our manuscript we have separated the limitations portion of the discussion section into its own section, found on page 7, lines 223-242

5.2. Please add some sentences to support why shorter median hospital lengths of stay was seen in the EI group in comparison to the IVP group.

Thank you for this feedback, we have included information on why we believe there were shorter hospital lengths of stay. This can be found on page 6, lines 190-194:

"This could be due to the longer % T>MIC achieved with EI infusion dosing, as mentioned previously, which in turn has been shown to denote better bacteriostatic and bactericidal properties [3,14], reductions in bacterial burden [17], and lengths of stay, especially ICU [14,18]."

Reviewer 3 Report

Comments and Suggestions for Authors

Thank you for this well designed pilot study concerning very actual subject of what is the best method of meropenem admition. Very high value is the real-world setting of the presented data, however my main concern is the randomisation method and it effectiveness, because of differences between both compared groups (time from admission to the first dose of antibiotic, elevated respiratory rate, frequency od atb administration and what is the most important in my opinion - differences in bacterias detected in cultures). In my opinion this issue should be carefully explained.

The authors write about several limitations of this pilot study, and the discussion is very informative. 

In my opinion this pilot study should be publicated and one should wait on the results of a larger group of patients. Congratulations to the authors.

Author Response

Thank you for this well designed pilot study concerning very actual subject of what is the best method of meropenem admition. Very high value is the real-world setting of the presented data, however my main concern is the randomisation method and it effectiveness, because of differences between both compared groups (time from admission to the first dose of antibiotic, elevated respiratory rate, frequency od atb administration and what is the most important in my opinion - differences in bacterias detected in cultures). In my opinion this issue should be carefully explained.

The authors write about several limitations of this pilot study, and the discussion is very informative. 

In my opinion this pilot study should be publicated and one should wait on the results of a larger group of patients. Congratulations to the authors.

Thank you for the very kind words and for your feedback, we hope the following addresses your concerns (found below). We have also expanded on the differences in time to first dose between IVP and EI administration along with difference in invasive oxygen between the two groups. This can be found in the Limitations section on page 7, lines 229-234: "Third, there was a numerically larger difference in time to first dose between the IVP and EI group. We suspect this difference was due to a transition to EI infusion after being on an alternative antibiotic regimen initially. However, we adjusted for this difference in the cox-proportional hazards model. There was also a numerically higher number of patients receiving invasive oxygen at the start of meropenem initiation in the EI compared to the IVP group, which was also adjusted for using the cox proportional hazard model."

  • Randomization model:
    • Random selection does not mean that patients were randomized, so that is why for the primary outcome we adjusted for covariates in the cox-proportional hazard model since we know there are limitations to potential confounding with a non-randomized study.
  • Timing in antibiotics:
    • We suspect that the later start time in meropenem start in the EI group was due to a transition to EI infusion after being on an alternate antibiotic initially. However we adjusted for this difference in the cox-proportional hazards model (stated in limitations).
  • Respiratory rate:
    • EI patients were sicker patients; however, we adjusted for the use of mechanical ventilation in the cox-proportional hazards model. (limitations – mention both this and timing of antibiotics and say we adjusted for these using the cox proportional hazards model)
  • Frequency:
    • Although there were significant differences in frequency of admission of meropenem in the two groups, >95% of patients were appropriately renally adjusted so they would be receiving the frequency that was appropriate for their renal function.
  • Organisms:
    • We are unsure of the exact reasoning; however, one possible reason could be that the EI orders were more likely utilized due to culture growth, so more directed therapy than empiric. However, there was not a statistically significant difference in most organisms besides coli (which if high MIC we would not have used meropenem). There were also many patients (nearly 50%) in the IV push group that did not have any culture collected.

Round 2

Reviewer 2 Report

Comments and Suggestions for Authors

Thank you for the revisions.